# The relationship between gut and nasopharyngeal microbiome composition can predict the severity of COVID-19

Benita Martin-Castaño[1,2†], Patricia Diez-Echave[2,3†], Jorge García-García[2,3,4*], Laura Hidalgo-García[2,3], Antonio Jesús Ruiz-Malagon[2,3], José Alberto Molina-Tijeras[2,3], María Jesús Rodríguez-Sojo[2,3], Anaïs Redruello-Romero[2], Margarita Martínez-Zaldívar[2,5], Emilio Mota[5], Fernando Cobo[6], Xando Díaz-Villamarin[7], Marta Alvarez-Estevez[2,4,8], Federico García[2,4,8], Concepción Morales-García[9], Silvia Merlos[9], Paula Garcia-Flores[9], Manuel Colmenero-Ruiz[2,10], José Hernández-Quero[2,11], Maria Nuñez[2,12,13], Maria Elena Rodriguez-Cabezas[2,3], Angel Carazo[2,4], Javier Martin[14], Rocio Moron[2,12*], Alba Rodríguez Nogales[2,3‡], Julio Galvez[2,3,15‡]

[1]Centro de Salud Las Gabias, Distrito Granada-Metropolitano, Granada, Spain; [2]Instituto de Investigación Biosanitaria de Granada (ibs.GRANADA), Granada, Spain; [3]Department of Pharmacology, Center for Biomedical Research (CIBM), University of Granada, Granada, Spain; [4]Servicio Microbiología, Hospital Universitario Clínico San Cecilio, Granada, Spain; [5]Centro de Salud "Salvador Caballero", Distrito Granada-Metropolitano, Granada, Spain; [6]Servicio Microbiología, Hospital Universitario Virgen de las Nieves, Granada, Spain; [7]Instituto de Investigación Biosanitaria, Granada, Spain; [8]CIBER de Enfermedades Infecciosas (CIBER-Infecc), Instituto de Salud Carlos III, Madrid, Spain; [9]Respiratory Medicine Department, Hospital Universitario Virgen de las Nieves, Granada, Spain; [10]Servicio de Medicina Intensiva, Hospital Universitario Clínico San Cecilio, Granada, Spain; [11]Servicio de Enfermedades Infecciosas, Hospital Universitario Clínico San Cecilio, Granada, Spain; [12]Servicio Farmacia Hospitalaria, Hospital Universitario Clínico San Cecilio, Granada, Spain; [13]CIBER de Epidemiología y Salud Pública (CIBER-ESP), Instituto de Salud Carlos III, Madrid, Spain; [14]Department of Cell Biology and Immunology, Institute of Parasitology and Biomedicine López-Neyra, CSIC, Granada, Spain; [15]CIBER de Enfermedades Hepáticas y Digestivas (CIBER-EHD), Instituto de Salud Carlos III, Madrid, Spain

**\*For correspondence:**
jgar11gar@gmail.com (JG-G);
rmoronr@gmail.com (RM)

†These authors contributed equally to this work
‡These authors also contributed equally to this work

**Competing interest:** The authors declare that no competing interests exist.

## eLife Assessment

This potentially **valuable** work characterizes the changes in the microbial composition of the nasal and faecal microbiomes in COVID-19 patients based on disease severity. This study enhances the understanding of COVID-19 severity predictors by identifying changes in bacterial species abundance in nasopharyngeal and fecal samples as a biomarker for predicting disease severity. The methods and statistics used appear to be **solid** and in line with the standards of the field.

**Abstract** Coronavirus disease 2019 (COVID-19) is a respiratory illness caused by severe acute respiratory syndrome coronavirus 2 (SARS-CoV-2) that displays great variability in clinical phenotype. Many factors have been described to be correlated with its severity, and microbiota could play a key role in the infection, progression, and outcome of the disease. SARS-CoV-2 infection has been

associated with nasopharyngeal and gut dysbiosis and higher abundance of opportunistic pathogens. To identify new prognostic markers for the disease, a multicentre prospective observational cohort study was carried out in COVID-19 patients divided into three cohorts based on symptomatology: mild (n = 24), moderate (n = 51), and severe/critical (n = 31). Faecal and nasopharyngeal samples were taken, and the microbiota was analysed. Linear discriminant analysis identified *Mycoplasma salivarium*, *Prevotella dentalis*, and *Haemophilus parainfluenzae* as biomarkers of severe COVID-19 in nasopharyngeal microbiota, while *Prevotella bivia* and *Prevotella timonensis* were defined in faecal microbiota. Additionally, a connection between faecal and nasopharyngeal microbiota was identified, with a significant ratio between *P. timonensis* (faeces) and *P. dentalis* and *M. salivarium* (nasopharyngeal) abundances found in critically ill patients. This ratio could serve as a novel prognostic tool for identifying severe COVID-19 cases.

## Introduction

Coronavirus disease 2019 (COVID-19) is a respiratory illness caused by severe acute respiratory syndrome coronavirus 2 (SARS-CoV-2). The data reported in November 2023 revealed that over 700 million people have been infected with the virus (*World Health Organization, 2023*). The appearance of mutations and variants of concern has caused several additional waves of infection and compromise the effectiveness of existing vaccines and antiviral drugs (*Harvey et al., 2021*). Moreover, SARS-CoV-2 infection has shown to cause long-term effects on human health, although the mechanisms are still poorly described (*Crook et al., 2021*). Even though most COVID-19 cases are mild, disease can be severe, resulting in hospitalization, respiratory failure, or even death (*World Health Organization, 2023*). Therefore, a remarkable feature of SARS-CoV-2 infection is the great variability in clinical phenotype among infected people. Many factors can correlate with COVID-19 disease severity, including age, gender, body mass index, previous comorbidities, immune responses, and genetics (*Yang et al., 2020*; *Ellinghaus et al., 2020*; *Bastard et al., 2020*). Of note, and unfortunately, the determinants of infection outcome and the pathogenic mechanisms are not completely understood yet.

SARS-CoV-2 primarily infects the respiratory tract by binding to angiotensin-converting enzyme 2 (ACE2) receptor (*Zhou et al., 2020b*). However, a growing body of evidence suggests that SARS-CoV-2 can also infect other organs since viral particles and nucleic acids have been found in different biological samples, like sputum, bronchoalveolar lavage fluid, faeces, blood, and urine (*Peng et al., 2020*; *Sun et al., 2020*). Thus, studies employing single-cell RNA sequencing have demonstrated that ACE2 is present in various organs and tissues, including the gastrointestinal tract, where ACE2 receptors have been reported to be highly expressed (*Zhang et al., 2020*). This supports several lines of evidence suggesting a substantial involvement of the gastrointestinal tract in the pathogenesis of the disease, including the ability of SARS-CoV-2 to infect and replicate in intestinal enterocytes (*Lamers et al., 2020*), or to increase expression of the viral entry receptor (ACE2 receptor) and several membrane-bound serine proteases (such as transmembrane protease serine 2 [TMPRSS2] and TMPRSS4) in intestinal epithelial cells (*Zang et al., 2020*).

In addition, reports have shown that SARS-CoV-2 infection can disrupt nasopharyngeal and intestinal microbiota composition and promote dysbiosis, reducing diversity and increasing the presence of opportunistic pathogens, including *Staphylococcus*, *Corynebacterium*, and *Acinetobacter* bacteria (*Candel et al., 2023*; *Ancona et al., 2023*), which can make patients more susceptible to secondary infections, rising morbidity and mortality (*De Bruyn et al., 2022*). Besides local changes in the respiratory tract, alterations in the distal gut microbiota have also been observed in SARS-CoV-2 infection (*Gu et al., 2020*; *Zuo et al., 2020*; *Wu et al., 2021*; *Cao et al., 2021*). Previous studies have evidenced a relevant connection between the microbiome in the nasopharynx and the gut, and preliminary data have suggested a bidirectional interaction that could play a role in the development of the immune response both in healthy and pathological conditions, including SARS-CoV-2 infection (*Mancabelli et al., 2023*). In this sense, dysbiosis has been associated not only with the severity of the disease but also with the recovery processes (*Candel et al., 2023*; *Cao et al., 2021*); however, little is understood of the link with the establishment of different symptomatic profiles in this condition, and to date, few studies have focused on the association between COVID-19 severity and the nasopharyngeal and faecal microbiota, being examined simultaneously.

**Table 1.** Clinical data description of enrolled patients.

Differences were displayed as: [a]p<0.05 severe vs. mild; [b]p<0.05 moderate vs. mild; [c]p<0.05 severe vs. moderate. ANOVA or Kruskal were employed for numerical variables and Fisher's test for categorical variables.

| | Mild (n = 24) | Moderate (n = 51) | Severe (n = 31) | p-Value[a] | p-Value[b] | p-Value[c] |
|---|---|---|---|---|---|---|
| Clinical variables | | | | | | |
| Mean age (SD), years | 43±12 | 54±14[b] | 62±11[a,c,] | 0.000001 | 0.0041 | 0.017 |
| Male, n (%) | 8 (33) | 24 (47)[b] | 22 (71)[a,c,] | 0.0071 | 0.032 | 0.041 |
| *Symptoms, n (%)* | | | | | | |
| Presence of dyspnoea | 6 (33) | 38 (75)[b] | 26 (84)[a] | 0.00018 | 0.001 | - |
| Presence of gastrointestinal alteration | 5 (26) | 17 (33) | 10 (33) | - | | |
| High respiratory rate | 1 (4) | 11 (22) | 19 (63)[a,c] | 0.00018 | - | 0.004 |
| Low sPO$_2$ | 6 (33) | 22 (43) | 22 (71)[a,c,] | 0.001 | - | 0.021 |
| High heart rate | 1 (4) | 14 (27)[b] | 17 (55)[a,c,] | 0.0008 | 0.02 | 0.01 |
| *Comorbidities, n (%)* | | | | | | |
| Obesity | 5 (26) | 14 (27) | 10 (32) | - | | |
| Diabetes | 4 (20) | 9 (18) | 8 (26) | - | | |
| Asthma | 3 (3) | 3 (6) | 2 (7) | - | | |
| Cardiomyopathy | 3 (3) | 3 (6) | 13 (42)[a,c,] | 0.02 | 0.001 | - |
| *Plasma determinations* | | | | | | |
| Mean lymphocytes (SD), 103/µL | 1.1±0.6 | 1.4±0.6 | 1.2±2.7 | - | | |
| Median neutrophils (IQR), 103/µL | 6 [5.5;6.6] | 6.4 [4.2;8.6] | 7.9 [5.4;10.9] | - | | |
| Mean platelets (SD), 103/µL | 329.6±8.5 | 257.4±115[b] | 276.4 ± 93[a] | 0.031 | 0.018 | - |
| Median D-dimer (IQR), mg/L | 0.39 [0.2;0.8] | 0.6 [0.3;1] | 1.6 [0.9;4.3][a,c,] | 0.0001 | - | 0.0001 |
| Median ferritin (IQR), ng/L | 157 [126;179] | 487 [274;1027][b] | 829 [488;1376][a] | 0.0001 | 0.0001 | - |
| Median C reactive protein (IQR), mg/L | 3.4 [2.6;4] | 18.2 [7.8;41.9][b] | 162 [65;210][a,c,] | 0.0001 | 0.0002 | 0.0001 |

Considering that the efficacy of the COVID-19 vaccines and antiviral drugs against SARS-CoV-2 is compromised with the emergence of mutations and new variants of the virus (*Harvey et al., 2021*), new therapeutic approaches and prognostic tools are still necessary. Therefore, the characterization of the nasopharyngeal and intestinal microbiome will allow to identify predictive biomarkers for the diagnosis and prognosis of the disease, as well as possible therapeutic targets in the management of SARS-CoV-2.

## Results
### Study patients characteristics

A total of 106 patients who had laboratory confirmation of SARS-CoV-2 infection were included in the present study. Of note, none of these patients reported another viral infection at the time of enrolment. Based on the clinical spectrum criteria reported in the COVID-19 treatment guidelines, patients were categorized into three groups: mild (24 patients), moderate illness that needed hospitalization in Respiratory Unit (51 patients), and severe symptomatology and admitted in the intensive care unit (ICU) (31 patients) (*Table 1*). The age of the patients significantly increased with the severity of the symptoms. Patients included in the severe group were significantly older than those with mild or moderate symptoms (p<0.05; ANOVA). Patient inclusion was carried out evenly in terms of gender as 52 women and 54 men were recruited; nevertheless, a gender-related impact on the clinical course of these patients was observed since males were predominantly in the group of patients

with severe symptoms when compared with the mild illness group (p<0.05; Fisher's test). Regarding the symptomatology, mild patients showed symptoms of a mild respiratory infection, including fever, cough, and headache. Also, 33% of mild patients displayed dyspnoea and low oxygen saturation, 26% reported gastrointestinal complaints (including stomach ache, digestive discomfort, and diarrhoea); and 4% reported high respiratory and heart rates. In contrast, moderate and severe patients showed higher frequencies of dyspnoea, low oxygen saturation, and increased respiratory or heart rates (p<0.05; Fisher's test). When the different comorbidities were considered, patients in the group with severe symptoms showed a higher percentage of cardiomyopathy compared to those from the mild or moderate cohorts (42% vs 3% and 6%, respectively; p<0.05; Fisher's test). However, no significant differences were found in the prevalence of the other pathologies among groups. Regarding blood-based biomarkers, platelets (p<0.05; ANOVA), D-dimer, ferritin, and C reactive protein (CRP) (p<0.05; Kruskal–Wallis) correlated with the severity of the symptoms, being the severe group significantly different.

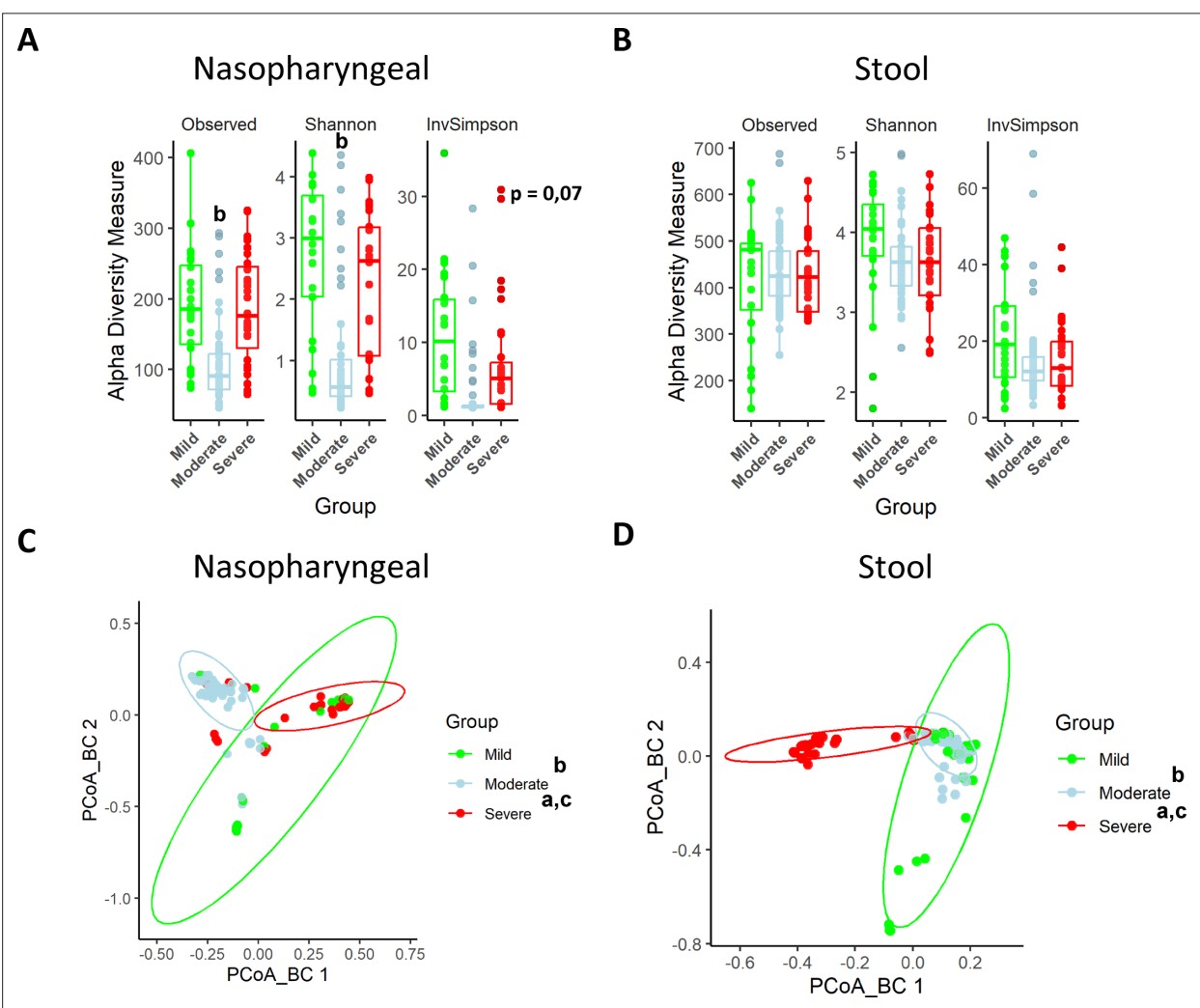

**Figure 1.** Nasopharyngeal and gut microbiota composition is modified depending on the severity of COVID-19 symptoms. (**A**) Alpha diversity index for nasopharyngeal swab samples microbiota. (**B**) Alpha diversity index for stool samples microbiota. (**C**) Principal Components Analysis (PCoA) for Bray–Curtis index of nasopharyngeal swab microbiota. (**D**) PCoA for Bray–Curtis index of stool samples microbiota. Values are represented as mean ± SD. Differences were displayed as: [a]p<0.05 severe vs. mild; [b]p<0.05 moderate vs. mild; [c]p<0.05 severe vs. moderate. PERMANOVA test was employed to determine Bray–Curtis significance differences.

## Bacterial composition differs between sample type and severity index in SARS-CoV-2-infected patients

Nasopharyngeal swabs and faeces were obtained from all the patients included in the study within 7 days of symptom onset and used for characterization of the microbiota composition. In the nasopharyngeal microbiota, α-diversity was reduced in the moderate and severe groups compared to the mild group (p<0.05 and p=0.07, respectively; ANOVA) (*Figure 1A*). Conversely, when α-diversity was examined in stool samples, no significant modifications were observed among groups (*Figure 1B*). On the other hand, ß-diversity analysis through the Bray–Curtis index revealed statistical differences between groups for both nasopharyngeal swabs and stools samples (p<0.001; PERMANOVA) (*Figure 1C and D*). As a result, microbiota is grouped in three separate clusters depending on the severity of the symptoms, which indicates that microbiota differences are based on changes in bacterial composition.

Similarly, the characterization of microbiota abundance composition revealed heterogeneity in the microbiota profile associated with severity and disease progression in these patients. Relative abundance levels for both phylum and genera were evaluated using the median abundance as a threshold. Specifically, genera with a median relative abundance greater than 0.5% were considered for detailed analysis.

At phylum level, in nasopharyngeal microbiota, the abundance of *Bacillota* was increased while the abundance of *Bacteroidota* and *Actinobacteroidota* was reduced in patients with severe symptomatology (*Figure 2A*). Conversely, the faecal microbiota of the three groups presented a more homogeneous distribution than the nasopharyngeal one, being the most abundant phyla *Bacillota* and *Bacteroidota* (*Figure 2B*). However, patients with severe symptoms, thus with a worse prognosis, showed a lower abundance of *Bacteroidota* (*Figure 2B*). Exact values of relative abundance for different phyla are shown in *Supplementary file 1*.

At genus level, the nasopharyngeal microbiome composition revealed that symptom severity was associated with a higher number of detected genera (*Figure 2C*). Of note, significant differences in genus abundance for each group of patients were found. Thus, mild patients presented a significantly higher abundance of *Alistipes*, *Muribaculaceae*, *Lactobacillus,* and *Lachnospiraceae* (p<0.05; Kruskal–Wallis), whereas the group with moderate symptoms showed a significant increase in *Alcaligenes*, *Pseudorobacter,* and *Pseudomonas* (p<0.05; Kruskal–Wallis). Additionally, patients with more severe disease had significantly higher relative abundance of *Acinetobacter*, *Actinomyces*, *Anaerococcus*, *Atopobium*, *Campylobacter*, *Dolosigranulum*, *Enterobacter*, *Enterococcus*, *Finegoldia*, *Fusobacterium*, *Gemella*, *Haemophilus*, *Lawsonella*, *Leptotrichia*, *Megasphaera*, *Neisseria*, *Serratia*, *Rotia,* and *Veillonella* (p<0.05; Kruskal–Wallis). Unlike nasopharyngeal swabs, stool samples showed a reduction in the number of detected genera as symptom severity increased (*Figure 2D*). Precisely, mild patients showed more presence of *Blautia*, *Muribaculaceae,* and different members of the *Clostridia* class (*Clostridia*, *Coprococcus,* and *Ruminococcus*) than the other groups (p<0.05; Kruskal–Wallis). In moderate ill patients *Bacteroides*, *Barneseilla*, *Faecalibacterium*, *Parabacteroides,* and *Streptococcus* were significantly increased (p<0.05; Kruskal–Wallis). Remarkably, *Alistipes*, *Anaerococcus*, *Dialister*, *Finelgoldia*, *Lachnocostridium*, *Prevotella,* or *Peptoniphilus* were more abundant in patients with severe symptoms (p<0.05; Kruskal–Wallis).

Relative abundance levels for genera identified bacteria and exact p-values can be found in *Supplementary files 2* and *3*.

## Differences in bacteria composition could be used as biomarkers to predict disease severity and outcome in SARS-CoV-2 infection

We investigated whether some specific taxa could be associated with the severity of the symptoms. Amplicon sequence variants (ASVs) were evaluated to determine core taxa along with the specific bacteria of each group of patients and samples (*Figure 3A and B*). Core taxa were identified using a detection threshold of 0.1 and a prevalence threshold of 0.1. In nasopharyngeal swabs, Venn diagram revealed that the three cohorts shared 51 core taxa. In total, 60 specific bacteria were identified in patients with mild symptoms while 32 were observed in patients with moderate symptoms and eight in patients with severe symptoms. When stools were considered, the three cohorts shared 159 core taxa. Here, 27 were specific of mild patients, another 33 of patients with moderate symptoms, and 27 of the severe patients (for more details, see *Supplementary file 4*).

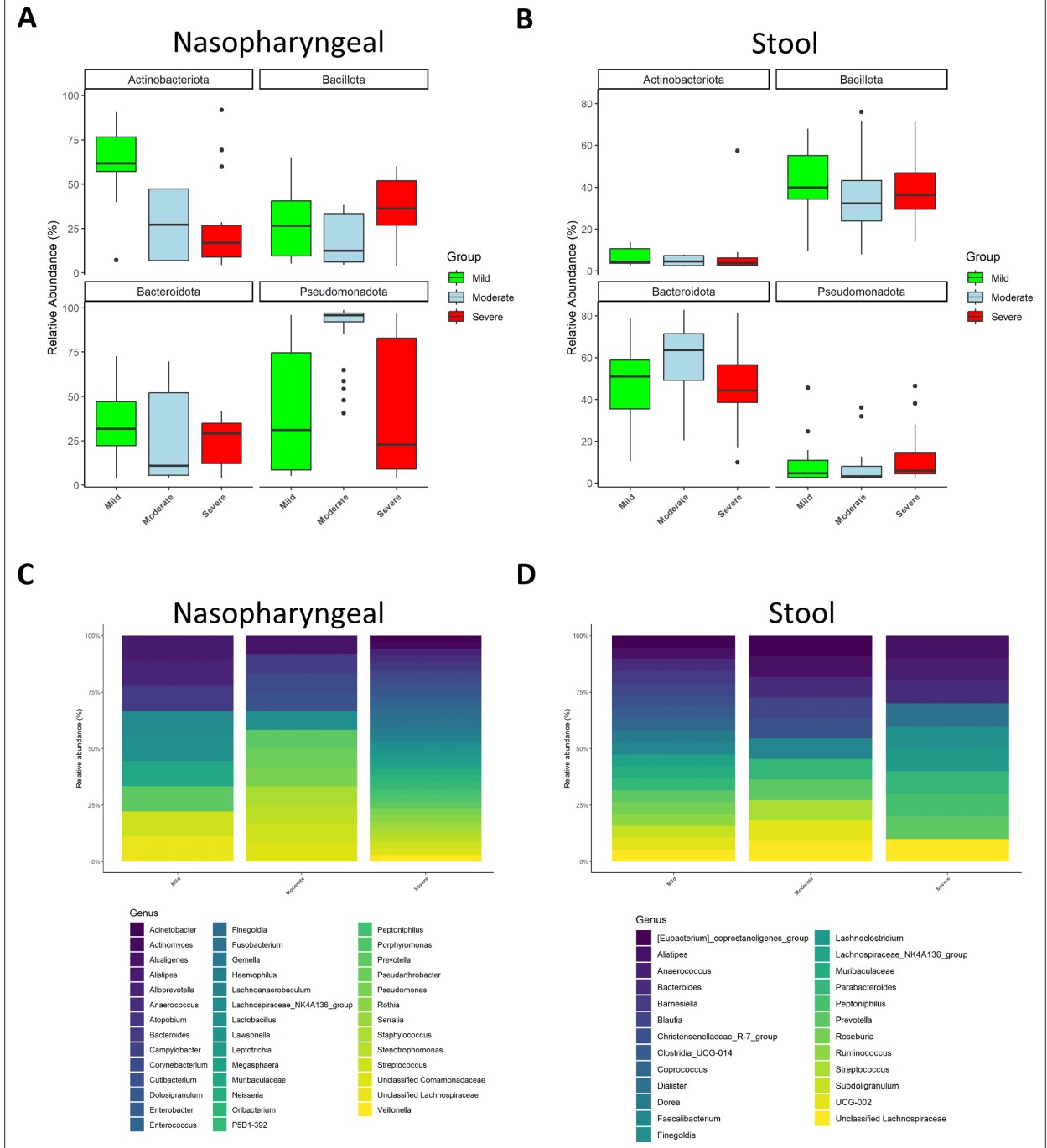

**Figure 2.** Microbiota composition of nasopharyngeal and stool samples at phylum level is slightly modified by COVID-19 symptoms severity. In contrast, at genus level, severity increases the total amount of detected bacteria in nasopharyngeal swabs while in stool samples it is promoting a reduction. (**A**) Representation of the relative abundance of the main phyla in nasopharyngeal swab samples. (**B**) Representation of the relative abundance of the main phyla in stool samples. (**C**) Relative abundance of the principal genera detected in nasopharyngeal swab samples. (**D**) Relative abundance of the principal genera detected in nasopharyngeal swab samples. For (**C**) and (**D**) only those amplicon sequence variants (ASVs) with a median higher to 0.5 were chosen.

Besides, the linear discriminant analysis (LEfSe) was performed to identify differential microorganisms for each group of patients (*Figure 3C and D*). In nasopharyngeal samples from mild patients, *Burkholderia* sp., *Paraburkholderia* sp., and *Massilia* sp. were determined; in moderate patients, *Pseudomonas veronii*, *Stenotrophomonas rhizophila*, and *Azotobacter chroococcum*; and in severe

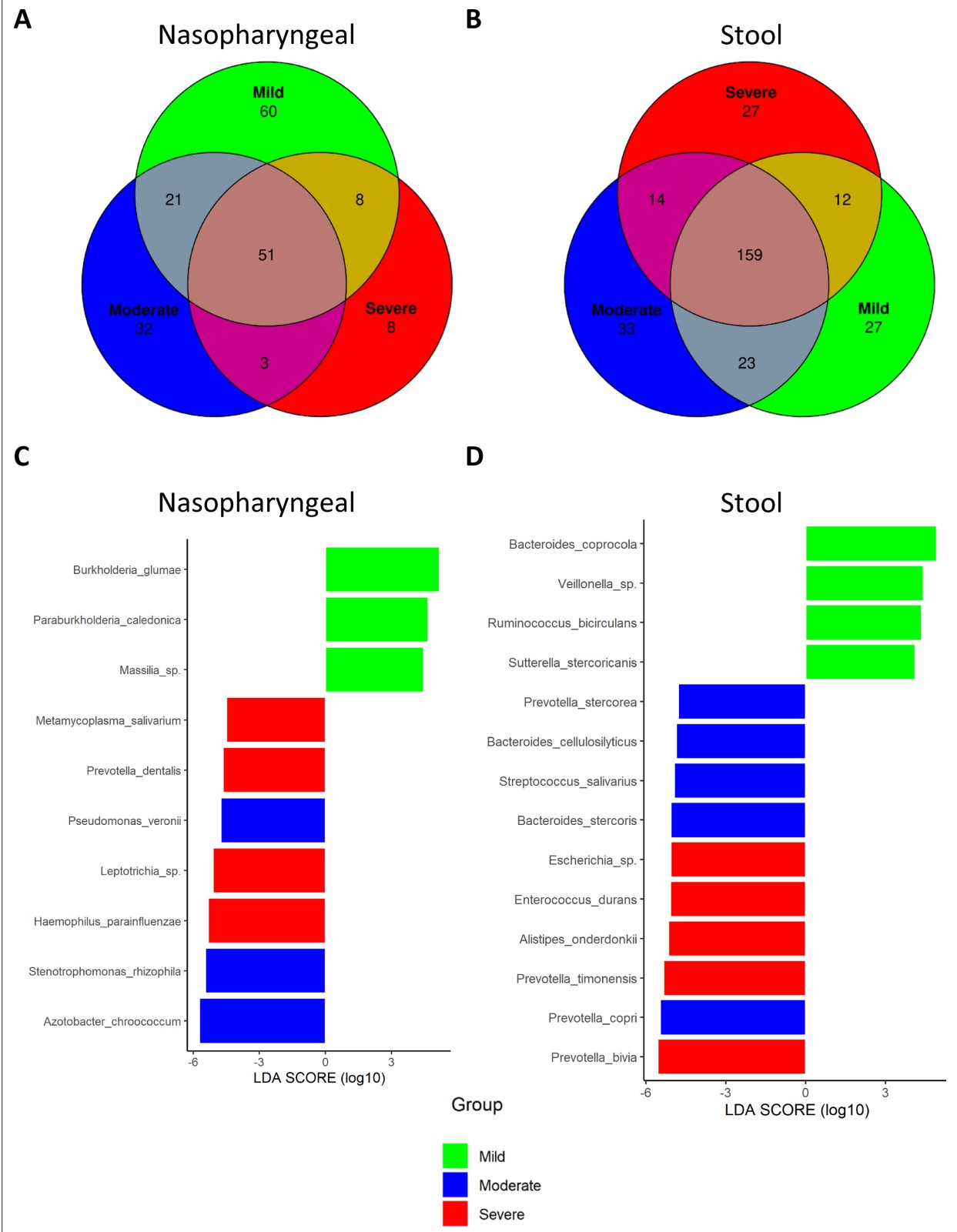

**Figure 3.** Differential analysis expression of microbiota composition from nasopharyngeal and stool samples revealed the presence of specific bacteria related to COVID-19 severity index. (**A**) Venn diagram showing amplicon sequence variants (ASVs) distribution in nasopharyngeal swab samples. (**B**) Venn diagram showing ASVs distribution in stool samples. (**C**) LEfSe plot of taxonomic biomarkers present in nasopharyngeal swab samples (p-value=0.01 and linear discriminant analysis [LDA] value = 4). (**D**) LEfSe plot of taxonomic biomarkers present in stool samples (p-value=0.01 and LDA value = 4). Venn diagrams were acquired with the following parameters: detection level = 0.01 and prevalence level = 0.01.

patients, *Mycoplasma salivarium*, *Prevotella dentalis*, *Leptotrichia,* and *Haemophilus parainfluenzae.* On the other hand, in stool samples, *Bacteroides coprocola*, *Veillonella* sp., *Ruminococcus bicirculans,* and *Sutterella stercoricanis* were identified as predictors of mild condition; *Prevotella stercorea*, *Bacteroides cellulosilyficus*, *Streptococcus salivarus*, *Bacteroides stercoris,* and *Prevotella copri* as predictors of moderate status; and *Escherichia*, *Enterococcus durans*, *Alistipes onderdonkii*, *Prevotella timonensis,* and *Prevotella bivia* as markers of severe condition. Unlike the Venn diagram, the presence of a species in the LEfSe analysis does not imply its absence in the other groups, but rather indicates significant differences in abundance. Hence, detailed abundance of these bacteria can be found in *Supplementary files 5* and *6*.

To further assess the role of these potential biomarkers in the prediction of COVID-19 severity, a correlation analysis was performed between these species and symptomatology. In summary, biomarkers identified in nasopharyngeal swabs in patients with severe condition, *M. salivarium* and *Leptotrichia,* showed a positive correlation with D-dimer and cardiomyopathy, respectively ($R > 0.3$). In addition, the other two biomarkers related to the highest severity, *H. parainfluenzae* and *P. dentalis,* also showed a positive association with CRP, D-dimer, and cardiomyopathy, even though it was not statistically significant ($R = 0.2$). More precisely, *M. salivarium* and *Leptotrichia* along with *P. dentalis* showed a significantly negative correlation towards lymphocytes count ($p<0.05$; $R < -0.25$) (*Figure 4A*). In the case of stool samples, biomarkers for mild symptomatology (*B. coprocola*, *R. bicirculans,* and *S. stercoricanis*) presented a negative correlation with CRP levels and respiratory rate ($p<0.05$; $R < -0.25$). In contrast, the severe biomarkers identified presented important correlations towards dyspnoea, cardiomyopathy, or respiratory rate. Concretely, *P. bivia* and *P. timonensis* revealed a positive correlation with D-dimer, ferritin levels, and respiratory rate, as well as a negative correlation with lymphocyte count ($R > 0.6$) (*Figure 4B*). Additionally, these two species showed a significant positive correlation against CRP levels and a negative correlation towards lymphocyte count ($p<0.05$). Exact correlation and p-values are shown in *Supplementary file 7*.

## Identification of a novel microbiome-based COVID-19 prognosis approach

Considering the possible identified biomarkers and their relationship towards clinical variables, we proposed a new approach to predict disease severity in patients suffering SARS-CoV-2 infection by evaluating correlation among specific bacteria abundance from nasopharyngeal and stool samples. The analysis revealed no important associations between nasopharyngeal and faecal microbiota in mild and moderate patients (*Figure 5A and B*). Nonetheless, in patients with severe symptoms the Spearman's rho coefficient showed a significant positive correlation between *P. timonensis* towards *P. dentalis* and *M. salivarium* (*Figure 5C*). Consequently, the ratios *P. timonensis/M. salivarium* and *P. timonensis/P. dentalis* were calculated to determine whether they could be used as a predictor of COVID-19 severity. Both seemed to be significantly increased in patients with severe symptoms compared to those with mild or moderate symptomatology ($p<0.05$; Kruskal–Wallis) (*Figure 5D and E*). As a result, the ratio between the abundance of these bacteria could serve as reliable predictors of severity of COVID-19.

## Discussion

Recent findings have evidenced the prominent role of the microbiome in several diseases, including those caused by viruses, proving that commensal microbiota could either enhance or hinder viral infections (*Mutlu et al., 2014*; *Guo et al., 2023*). A large body of evidence indicates that respiratory virus infections can induce a microbial imbalance in the airways, increasing the risk of secondary bacterial infections and worsening the prognosis (*Li et al., 2023*). Besides, alterations in the gut microbiota have also been noted during respiratory virus infections, likely influenced by the 'gut-lung axis' (*Budden et al., 2017*). In this regard, different studies have tried to establish relationships between microbiota composition and SARS-CoV-2 infection (*Yamamoto et al., 2021*; *Di Stadio et al., 2020*) but just a few have explored microbiota compositional variations among patients with different symptomatology (*Nguyen et al., 2023*). However, only a few studies have explored the nasopharyngeal-faecal axis as a potential biomarker of severity in patients infected with SARS-CoV-2. Hence, the

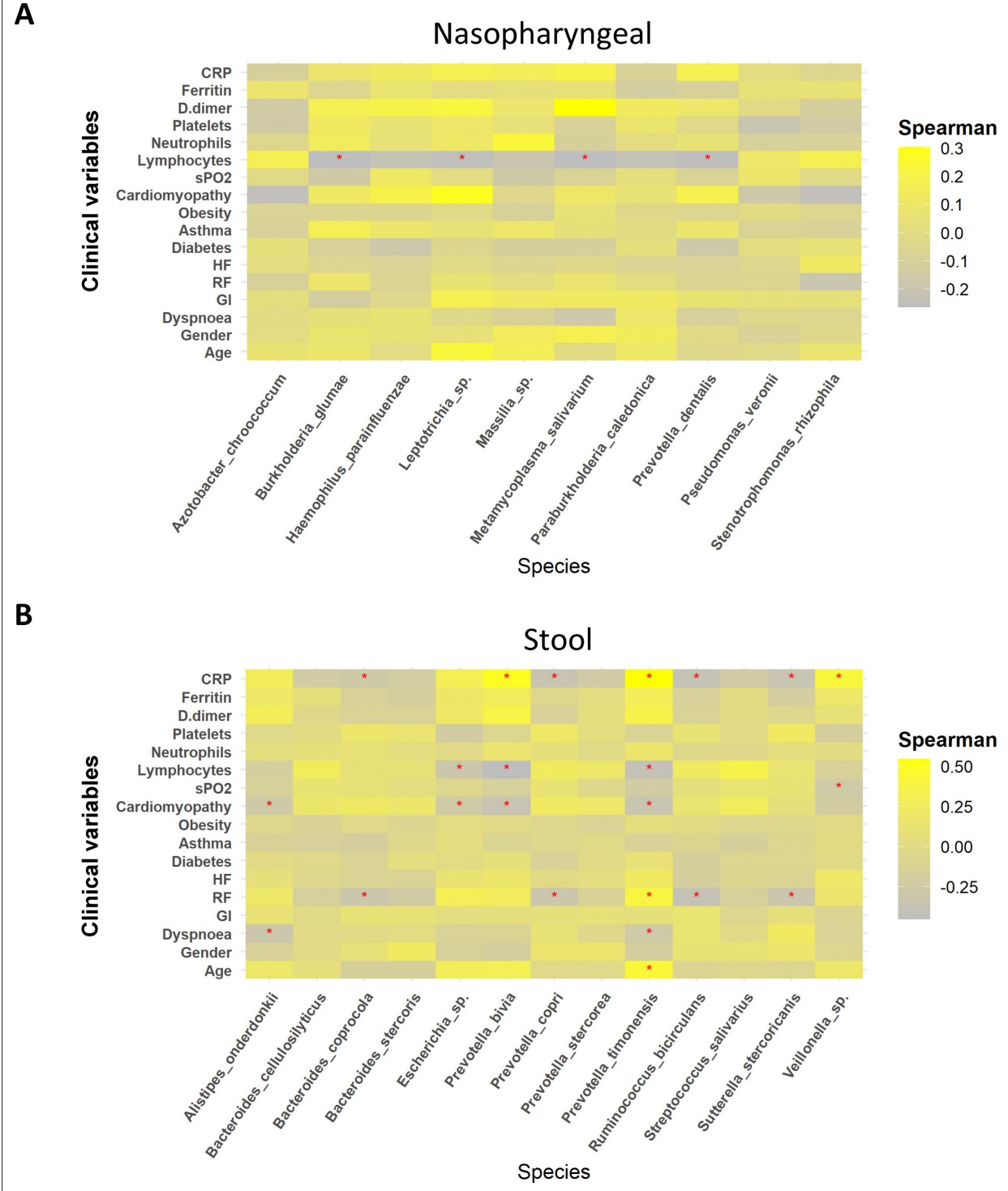

**Figure 4.** Whereas mild biomarkers showed negative correlations towards clinical variables, severe biomarkers presented positive correlations. (**A**) Correlation plot of nasopharyngeal swab biomarkers and clinical variables. (**B**) Correlation plot of stool samples biomarkers and clinical variables. RR: respiratory rate; HR: heart rate; GI: gastrointestinal alterations. Correlation was calculated by taking into account abundance levels of each bacteria against the clinical variable of severe ill patients. Red asterisks stand for p-values<0.05 and correlations ≥0.3 or ≤–0.25.

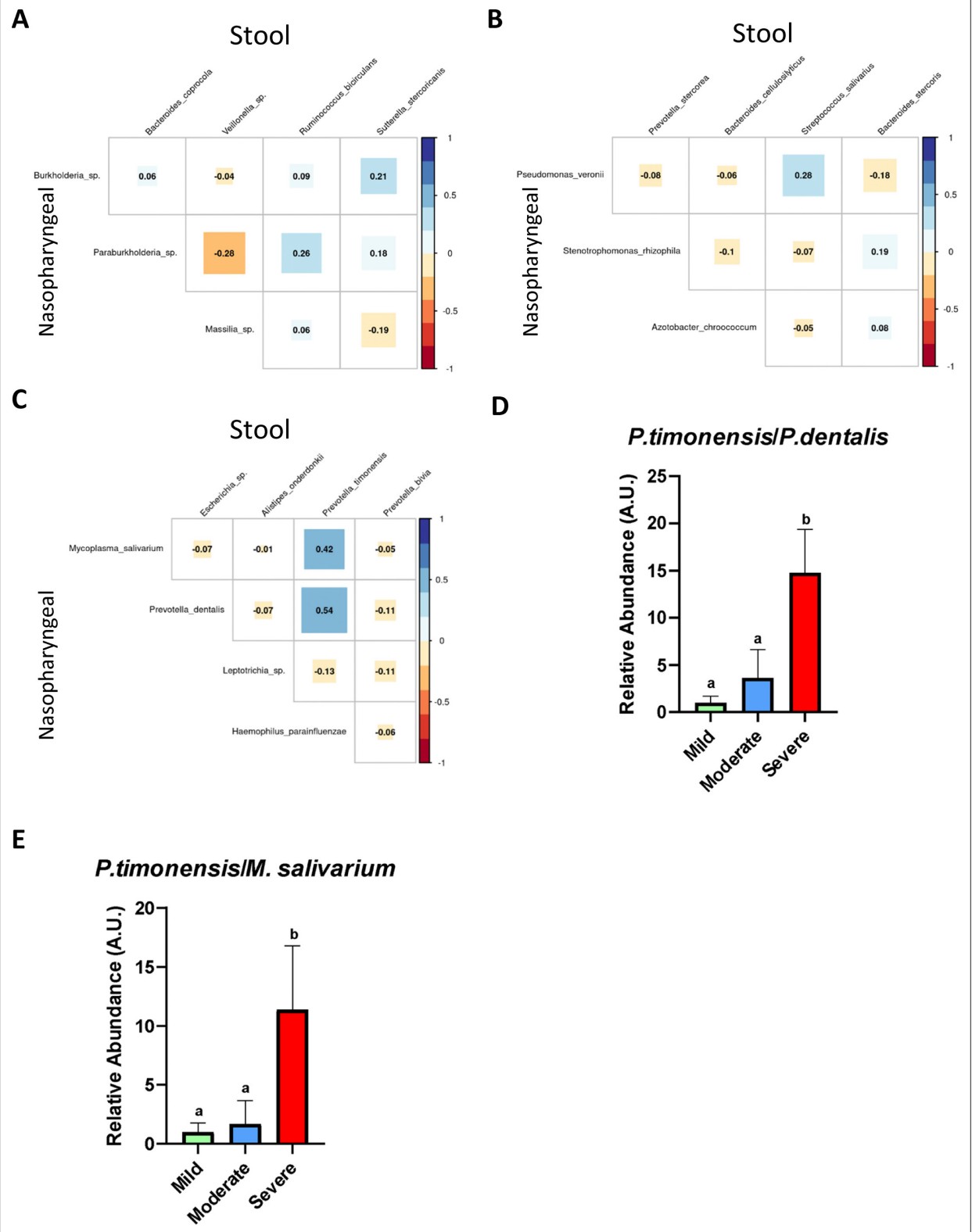

**Figure 5.** The existence of a relationship between the abundance of nasopharyngeal severe biomarkers and stool severe biomarkers allows the employment of an abundance ratio between them as a new tool for predicting COVID-19 severity. (**A**) Correlation plot among biomarkers found in nasopharyngeal swab and stool samples in mild condition. (**B**) Correlation plot among biomarkers found in nasopharyngeal swab and stool samples in moderate condition. (**C**) Correlation plot among biomarkers found in nasopharyngeal swab and stool samples in severe condition. (**D**) Ratio of the abundance between *P. timonensis* (stool) and *M. salivarium* biomarkers. (**E**) Ratio of the abundance between *P. timonensis* (stool) and *P.dentalis* (nasopharyngeal swab) biomarkers. [a]p<0.05 severe vs. mild; [b]p<0.05 moderate vs. mild.

present study provides some biomarkers located in nasopharyngeal and gut microbiota that could help to predict COVID-19 severity.

In this context, 106 patients with SARS-CoV-2 infection were included in this study and classified into three severity groups according to their symptoms. To ensure results were derived only from SARS-CoV-2 infection, included patients were not positive for other viral infections. Considering clinical parameters, the current study confirmed that age and gender are positively associated with more severe symptoms like dyspnoea, increased heart, and respiratory rates together with lower oxygen saturation. Results that are in line with other COVID-19 studies conducted globally have shown that men and women are disproportionately affected. Data revealed that males suffered from more severe disease than females, including higher ICU admission rates, dyspnoea, increased heart rate, being all clinical signs for higher severity of the COVID-19 disease (*Pijls et al., 2022*; *Mauvais-Jarvis, 2020*). Regarding some biochemical parameters those groups of patients with moderate or severe symptoms showed significantly increased levels of D-dimer, ferritin, and CRP, which are typically described as biomarkers for COVID-19 severity (*Gómez-Pastora et al., 2020*; *Rostami and Mansouritorghabeh, 2020*; *Zhou et al., 2020a*). Interestingly, previous studies have reported that alterations in the gut microbiota composition in COVID-19 patients could be also associated with disease severity, measured with the elevated concentration of the blood markers described above (*Liu et al., 2022a*; *Liu et al., 2022b*; *Yeoh et al., 2021*). In this sense, we have confirmed these results in our study as specific bacteria from severe condition positively correlate with the levels of ferritin, CRP, D-dimer, or lymphocytes. Consequently, the microbiota could be key in the clinical phenotype of COVID-19 patients, although its specific contribution to the progression of the infection and a poor prognosis needs to be better understood.

The results of alpha diversity revealed only substantial changes in richness in the nasopharyngeal microbiome associated with moderate and severe symptoms. Although most of the studies have proposed that SARS-CoV-2 infection is associated with lower microbial diversity in nasopharyngeal samples (*Candel et al., 2023*; *Gao et al., 2021*; *Gupta et al., 2022*), others did not find differences in alpha diversity composition among groups with different symptomatology (*De Maio et al., 2020*; *Braun et al., 2021*). This could be explained by the diverse methods of sample collection, and also the SARS-CoV-2 variant and the treatment with antibiotics (*Bose et al., 2023*).

Furthermore, in the present study, beta diversity analysis showed that in both types of samples every group of patients presented distinctly differentiated clusters. As previously reported, COVID-19 disease severity is more dependent on the presence or absence of certain bacteria rather than alterations in bacterial diversity and richness (*Wang et al., 2023*). This argument was supported by the characterization of bacterial microbiota composition at phyla and genera levels. In nasopharyngeal swabs, the abundance of *Bacteroidota* and *Actinobacteriota* was reduced compared to mild patients, which goes in accordance with previous reports where these bacteria have been previously linked to a better prognosis of SARS-CoV-2 infection since it can exert beneficial effects by preventing respiratory diseases, including COVID-19 (*Nardelli et al., 2022*; *Bozkurt and Bilen, 2021*; *Wei et al., 2023*; *Bassis et al., 2014*; *Pérez-Losada et al., 2023*). Moreover, the abundance of *Bacillota* and *Pesudomonadota* was increased in patients with severe symptomatology, thus supporting previous studies in which higher counts of *Bacillota* (*Staphylococcus* sp. and *Streptococcus* sp.) and *Pesudomonadota* (*Pseudomonas* sp.) were associated with moderate and severe symptoms of COVID-19 (*Garcia-Vidal et al., 2021*).

Regarding genera composition, *Alistipes* and *Muribaculaceae* were highly abundant in patients with mild symptoms. These bacteria have been well characterized in gut microbiota, but little is known about their presence in nasopharyngeal microbiota. Different experimental studies in mice have suggested their role in viral infections. Thus, *Muribaculaceae* was found in the lung microbiota in SARS-CoV-2-infected mice that were treated with a selective inhibitor of the main protease (M^pro) (*Seibert et al., 2021*). In the case of *Alistipes,* in a study conducted in children infected with respiratory syncytial virus (RVS), its abundance was reduced in the nasopharyngeal microbiota in comparison with non infected subjects (*Schippa et al., 2020*). Overall, these genera could be associated with a protective role against viral infection, and their higher presence in nasopharyngeal samples from mild COVID-19 patients could prevent the progression to severe disease.

In contrast, the increased presence of other genera, such as *Corynebacterium*, *Acinetobacter*, *Staphylococcus,* and *Veillonella*, was associated with the severity of SARS-CoV-2 infection. This is

supported by previous studies in which these genera were associated with both disease severity and systemic inflammation (*Ma et al., 2021*). In addition, higher abundance of *Enterococcus* was observed in severe patients, thus confirming other studies in critically ill patients (*Merenstein et al., 2021*).

On the contrary, findings at phylum level in stool samples did not reveal notable modifications among patient groups. However, in mild ill patients, different genera from *Clostridia* class (*Clostridia*, *Coprococcus*, *Dorea*, *Lachnospiraceae*, *Roseburia,* and *Ruminococcus*) and *Barnesiella* were identified as highly abundant. While the class *Clostridia* was associated with a reduced production of proinflammatory cytokines in COVID-19 patients and in those who recovered from the infection (*Mizutani et al., 2022*; *Mańkowska-Wierzbicka et al., 2023*), *Barnesiella* prevents colonization by antibiotic-resistant bacteria such as *Enterococcus*, which has been reported as a frequent cause of systemic infection in critically ill COVID-19 patients (*Ubeda et al., 2013*; *Giacobbe et al., 2021*). Thus, the reduction of gut abundance of *Barnesiella* and *Clostridia* members could be associated with more severe symptoms in COVID-19 patients. Moreover, the specific genera detected in the severe illness group, *Lachnocostridum*, *Anaerococcus,* and *Peptoniphilus*, have been previously recognized as opportunistic pathogens, which could induce gut inflammation and contribute to a poor prognosis (*Murphy and Frick, 2013*).

Considering the variations observed in both nasopharyngeal and gut microbiota composition and their miscellaneous effects, the identification of specific bacteria could be used as a biomarker to predict COVID-19 severity. Accordingly, the present study has identified unique ASVs for the different severities of COVID-19. For nasopharyngeal samples, species belonging to the genus *Lactobacillus* (*L. fermentum* or *L. reuteri*) or *Prevotella* (*P. pallens, P. ori,* and *P. shahii*) have been pinpointed. Interestingly, mild patients have also shown a higher content of *Lactobacillus*, which could also contribute to the host defence against viral infection at early stages, as reported in asymptomatic COVID-19 patients (*Kageyama et al., 2022*). In the case of *Prevotella* sp., its role in COVID-19 infection has not been clearly elucidated, but previously published microbiome analysis has revealed that its abundance was higher in mild patients (*Chen et al., 2022*). Nevertheless, others have suggested that it could be a biomarker of critical phenotype in COVID-19 patients (*Haran et al., 2021*; *Lu et al., 2023*). It is interesting to highlight that *Anaerococcus prevotii* was one of the species exclusively found in stools in mild patients. This microorganism has already been linked to lower inflammation in COVID-19 patients (*Seong et al., 2023*). Conversely, *Coprobacillus cateniformis* was only found in severe patients, and could be involved in the development of a worse condition in these patients through ACE2 upregulation (*Zuo et al., 2020*).

Even though these bacteria were found to be unique for each group, LEfSe was performed to obtain specific severity prognostic biomarkers depending on the abundance threshold. Therefore, for mild patients, *Burkholderia* and *Paraburkholderia* were identified in nasopharyngeal swabs and *B. coprocola* and *R. bicirculans* in stool samples. Although the information regarding the first two species in humans is limited, a few studies have reported their presence in the commensal human microbiota (*Ning et al., 2022*; *Mengyi et al., 2023*). Contrarily, *B. coprocola* and *R. bicirculans* have been found in both healthy and COVID-19 patients, being the abundance of *R. bicirculans* reduced in infected subjects (*Li et al., 2021*).

Correspondingly, in patients with moderate symptoms, *P. veronii* was detected in nasopharyngeal samples whereas *P. stercorea*, *B. cellulosilyficus*, *B. stercoris,* and *P. copri* were identified in stool samples. In general, these findings agree with previous studies (*de Nies et al., 2023*). Xu et al. found that infected patients showed higher abundance of *B. cellulosilyficus* (*Xu et al., 2022*), whereas *B. stercoris* and *P. copri* were associated with ACE2 upregulation and increased proinflammatory cytokine production, respectively, in COVID-19 patients (*Li et al., 2021*; *Lymberopoulos et al., 2022*).

Likewise, in critically ill patients, the biomarkers detected for nasopharyngeal microbiota were *M. salivarium*, *P. dentalis*, *Leptotrichia,* and *H. parainfluenzae*, while, in stool samples, *Escherichia* sp., *E. durans*, *A. Onderdonkii*, *P. timonensis,* and *P. bivia* were the species recognized as biomarkers. Both *P. bivia* and *P. timonensis* have been defined as unique microorganisms in COVID-19 patients' microbiota (*Li et al., 2021*; *Thissen et al., 2022*), whilst a higher abundance of *M. salivarium*, *H. parainfluenzae,* and *E. durans* were related to poor prognosis (*Sulaiman et al., 2021*; *Devi et al., 2023*; *DeVoe et al., 2022*). When considering *A. onderdonkii*, it is a short-chain fatty acid-producing bacteria that has been reported to be inversely associated with COVID-19 severity (*Zuo et al., 2020*; *Wang et al., 2023*); however, there is conflicting evidence regarding its pathogenicity that indicates

that *A. onderdonkii* may have protective effects against some diseases, including COVID-19 (*Zuo et al., 2020*). Nonetheless, considering that Zuo et al. employed a different methodology to analyse microbiota composition from stool samples, *A. onderdonkii* could be considered as a biomarker of severe condition in SARS-CoV-2-infected subjects.

Of note, the use of these bacteria as biomarkers of severity in SARS-CoV-2 infection is further supported by the fact that these species exhibited positive correlations with critical clinical variables mentioned above. Specifically, possible severe biomarkers from nasopharyngeal samples such as *M. salivarium*, *H. parainfluenzae,* and *P. dentalis* showed a negative correlation towards lymphocytes count. This also happens for stool severe biomarkers *P. bivia* and *P. timonensis*. As it is well established that microbiota can modify blood cells count (*Khosravi et al., 2014*), it would not be a surprise that these bacteria could promote COVID-19 severity by lowering lymphocyte count. In addition, *P. bivia* and *P. timonensis* showed positive correlation with ferritin, CRP, and D-dimer levels, as well as cardiomyopathy and respiratory rates. Several studies have revealed both the relationship between CRP levels, gut microbiota, and COVID-19 severity (*Moreira-Rosário et al., 2021*), as well as the positive correlation of specific bacteria with D-dimer, CRP, and the levels of pro-inflammatory mediators in plasma (*Zhou et al., 2021*).

Finally, although the composition of the nasal and faecal microbiota in COVID-19 patients has been previously studied (*Li et al., 2023*), the interaction between these microbiotas has not been extensively evaluated. In this study, we analysed the relationship between specific bacteria from nasopharyngeal and stool samples, confirming a correlation between the abundances of species in these different sample types. Concretely, a strong positive correlation between *P. timonensis* (in stool samples) towards *P. dentalis* and *M. salivarium* (in nasopharyngeal samples) was found in severe patients. To maximize the potential use of these biomarkers, the ratio of the abundance of these species was also significantly increased within the highest severity of this condition. As a result, the ratio proposed in this study could be used as a novel predictor to identify critically ill COVID-19 patients as the ratio *Bacillota* and *Bacteroidetes* has been used as a marker of dysbiosis (*Ley et al., 2006*). In this case, the ratio *P. timonensis/P. dentalis* and *P. timonensis/M. Salivarium* could be a prognostic tool for severe SARS-CoV-2, and its increase could be associated with a higher risk COVID-19 severity development.

## Conclusion

This inter-individual variability between the COVID-19 patients could contribute to the different symptomatology observed. This study has identified a correlation between changes in the nasopharyngeal and stool microbiota with COVID-19 severity. A novel biomarker linked to severity of COVID-19 infection has been described based on changes in the abundance of bacterial species in nasopharyngeal and faecal samples. This knowledge can support the development of biomarkers to gauge disease severity and, also, the design of novel therapeutic strategies to mitigate adverse outcomes. Further investigations are imperative to explore how the association between nasopharyngeal and faecal microbiota can be modulated to uncover its role in enhancing immune health, preventing or treating SARS-CoV-2 infections, and fostering immunity.

## Materials and methods
### Subject recruitment and sample collection

A multicentre prospective observational cohort study was carried out between September 2020 and July 2021. Patients with SARS-CoV-2 infection were recruited from the University Hospital San Cecilio, the University Hospital Virgen de las Nieves, and the Primary Care centres, Salvador Caballero and Las Gabias in Granada (Spain). These patients were Caucasian and laboratory-confirmed SARS-CoV-2 positive by quantitative reverse transcription PCR (RT-qPCR) performed on nasopharyngeal swabs collected by healthcare practitioners. No confounding factors such as other viral infections were present. Patients were classified into three groups based on severity profile following the described guidelines (*covid19treatmentguidelines, 2019*); mild (n = 24), moderate (n = 51), and severe/critical (n = 31). Mild illness includes individuals who have any of the various signs and symptoms of COVID-19 (e.g. fever, cough, sore throat, malaise, headache, muscle pain, nausea, vomiting, diarrhoea, loss of taste and smell) but do not have shortness of breath, dyspnoea, or abnormal chest imaging. Moderate illness includes patients that show fever and respiratory symptoms with

radiological findings of pneumonia. Severe/critical illness includes patients with any of the following criteria: respiratory distress (≥30 breaths/min), oxygen saturation ≤93% at rest, arterial partial pressure of oxygen (PaO$_2$)/fraction of inspired oxygen (FiO$_2$) ≤ 300 mmHg, respiratory failure and requiring mechanical ventilation, shock, and with other organ failures that required ICU care.

Nasopharyngeal swabs and stools samples from patients were collected by healthcare staff in the moderate and severe/critical cohorts, while the patients of the mild cohort provided stools self-sampled at home. Stools and nasopharyngeal swabs were collected in collection tubes containing preservative media (OMNIgene•GUT, DNAGENOTEK, Ottawa, Ontario, Canada) and stored at −80°C until processing.

## Microbial DNA extraction, library preparation, and next-generation sequencing

For all faecal and nasopharyngeal samples, DNA was isolated according to the modified protocol reported by *Rodríguez-Nogales et al., 2017* using QIAGEN Allprep PowerFecal DNA kit (QIAGEN, Germany). DNA was quantified using Qubit dsDNA HS assay kit (12640ES60, Yeason Biotechnology, Shanghai, China) and total DNA was amplified by targeting variable regions V3-V4 of the bacterial 16S rRNA gene. Quality control of amplified products was achieved by running a high-throughput Invitrogen 96-well-E-gel (Thermo Fisher Scientific, Waltham, MA). PCR products from the same samples were pooled in one plate and normalized with the high-throughput Invitrogen SequalPrep 96-well Plate kit. Pool samples were sequenced using an Illumina MiSeq.

## Bioinformatic tools

Paired end reads quality was checked with FastQC software (*Bittencourt, 2010*). Trimming of adapters and filtering of low-quality sequences was performed with Trimmomatic (*Bolger et al., 2014*). Filtered reads were further processed with QIIME2 software (open access, Northern Arizona University, Flagstaff, AZ) by employing DADA2 software (*Callahan et al., 2016*) to carry out denoising steps to obtain ASVs. SILVA reference database (138 99% full length) was used for taxonomic assignment (*Kaehler et al., 2019*). Microbiota results were further analyse with the help of different R packages. Alpha and beta diversity as well as relative abundance were appraised with the *Phyloseq* package (*Love et al., 2014*). Beta diversity differences were obtained with a Permutational Multivariate Analysis of Variance (PERMANOVA) included in the *Vegan* package *Eulerr* along with *microbiomeutilities* package [*Sudarshan and Shetty, ., 2020*] was utilized for constructing Venn diagrams [*R Development Core Team, 2013*]. Meanwhile, *microbial* package was employed to perform linear discriminant analysis (LDA) effect size (LEfSe) with an LDA score of 3 (*R Development Core Team, 2013*).

## Statistical analysis

All data presented were analysed and visualized using R software. For numerical clinical variables, normality and homoscedasticity were assessed using the shapiro.test and leveneTest functions from the *stats* and *car* package, respectively. Data were displayed as mean ± SD for normally distributed variables, whereas median and interquartile range (IQR) were used for variables with non-normal distributions.

Statistical differences were evaluated using ANOVA for parametric data or Kruskal–Wallis test for non-parametric data, both available in the stats package. When significant differences were identified, post hoc group comparisons were conducted using Tukey's method through the TukeyHSD function for ANOVA and Dunn's test via the dunnTest function from the FSA package for Kruskal–Wallis.

Categorical variables were expressed as percentages, and statistical significance was determined using Fisher's exact test with the fisher.test function from the *stats* package. Finally, for correlation analysis, Spearman's correlation coefficient was employed for non-normally distributed variables, while Pearson's correlation coefficient was used for normally distributed variables. The cor function from the *stats* package was used to calculate these coefficients, and the *corrplot* package was employed for visualization. Statistical significance was defined as a p-value of <0.05. To illustrate group differences, different letters (a, b, or c) were used to denote statistically significant differences among groups. All data, models, and analytical output are on the linked GitHub repository https://github.com/albarodnog/COVID-INMUNOBIOTICS-Group copy archived at *Rodríguez Nogales, 2024*.

## Acknowledgements

We acknowledge the collaboration of all the participants who voluntarily and selflessly participated in the study. The research project was supported by Government of Andalucia (Spain) (CV20-99908).

## Additional information

### Funding

| Funder | Grant reference number | Author |
|---|---|---|
| Junta de Andalucía | CV20-99908 | Julio Galvez |

The funders had no role in study design, data collection and interpretation, or the decision to submit the work for publication.

### Author contributions

Benita Martin-Castaño, Conceptualization, Investigation, Methodology; Patricia Diez-Echave, Laura Hidalgo-García, Antonio Jesús Ruiz-Malagon, José Alberto Molina-Tijeras, María Jesús Rodríguez-Sojo, Investigation, Methodology; Jorge García-García, Data curation, Software, Formal analysis, Investigation, Methodology, Writing – original draft, Writing – review and editing; Anaïs Redruello-Romero, Methodology; Margarita Martínez-Zaldívar, Resources, Data curation; Emilio Mota, Fernando Cobo, Data curation; Xando Díaz-Villamarin, Marta Alvarez-Estevez, Concepción Morales-García, Silvia Merlos, Paula Garcia-Flores, José Hernández-Quero, Maria Nuñez, Investigation; Federico García, Manuel Colmenero-Ruiz, Resources, Investigation; Maria Elena Rodriguez-Cabezas, Investigation, Writing – review and editing; Angel Carazo, Javier Martin, Validation, Investigation; Rocio Moron, Formal analysis, Supervision, Validation, Investigation, Methodology, Writing – original draft, Writing – review and editing; Alba Rodríguez Nogales, Formal analysis, Supervision, Validation, Investigation, Writing – original draft, Writing – review and editing; Julio Galvez, Conceptualization, Resources, Formal analysis, Supervision, Funding acquisition, Validation, Visualization, Writing – original draft, Writing – review and editing

### Author ORCIDs

Jorge García-García ⓘ https://orcid.org/0000-0003-0925-9441
Xando Díaz-Villamarin ⓘ https://orcid.org/0000-0003-0849-9902
Alba Rodríguez Nogales ⓘ https://orcid.org/0000-0003-1927-0628
Julio Galvez ⓘ https://orcid.org/0000-0001-6876-3782

### Ethics

The study was conducted in accordance with the declaration of Helsinki and the protocol approved by the Clinical Research Ethics Committee of Granada (CEIC) (ID of the approval omicovid-19 1133-N-20). All patients provided written informed consent before being included in the study. The samples were managed by the ibs. GRANADA Biobank following the protocols approved by the Andalusian Biomedical Research Ethics Coordinating Committee.

Reviewer #1 (Public review): https://doi.org/10.7554/eLife.95292.3.sa1
Reviewer #3 (Public review): https://doi.org/10.7554/eLife.95292.3.sa2
Author response https://doi.org/10.7554/eLife.95292.3.sa3

## Additional files

### Supplementary files

Supplementary file 1. Relative abundance of phyla found in nasopharyngeal and stool samples.

Supplementary file 2. Relative abundance and exact p-values of genera found in nasopharyngeal samples. [a] $p<0.05$ severe vs. mild; [b] $p<0.05$ moderate vs. mild; [c] $p<0.05$ severe vs. moderate. ANOVA or Kruskal were employed for numerical variables and Fisher's test for categorical variables. '-' means that genera were no present in microbiota from that group or that not significant differences

were found.

Supplementary file 3. Relative abundance and exact p-values of genera found in stool samples. [a] p<0.05 severe vs. mild; [b] p<0.05 moderate vs. mild; [c] p<0.05 severe vs. moderate. ANOVA or Kruskal were employed for numerical variables and Fisher's test for categorical variables. '-' means that genera were no present in microbiota from that group or that not significant differences were found.

Supplementary file 4. Unique ASVs identified for each group in each sample.

Supplementary file 5. Relative abundance of possible biomarkers of stool samples.

Supplementary file 6. Relative abundance of possible biomarkers of nasal swab samples.

Supplementary file 7. Correlation values and exact p-values of the relationship between microbiota and clinical variables.

MDAR checklist

### Data availability

Sequencing data have been deposited in Harvard Database (16S DNA sequences from stool samples of patients infected by SARS-CoV-2). Code is available in a GitHub repository (https://github.com/albarodnog/COVID-INMUNOBIOTICS-Group copy archived at *Rodríguez Nogales, 2024*).

The following dataset was generated:

| Author(s) | Year | Dataset title | Dataset URL | Database and Identifier |
|---|---|---|---|---|
| Jorge G | 2025 | 16S DNA sequences from stool samples of patients infected by SARS-CoV-2 | https://doi.org/10.7910/DVN/ZYRTSO | Harvard Dataverse, 10.7910/DVN/ZYRTSO |

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
