## [Editor Report · eLife Assessment]

This potentially **valuable** work characterizes the changes in the microbial composition of the nasal and faecal microbiomes in COVID-19 patients based on disease severity. This study enhances the understanding of COVID-19 severity predictors by identifying changes in bacterial species abundance in nasopharyngeal and fecal samples as a biomarker for predicting disease severity. The methods and statistics used appear to be **solid** and in line with the standards of the field.

---

## [Referee Report · Reviewer #1 (Public review)]

Summary:

The research study under review investigated the relationship between gut and identified potential biomarkers derived from the nasopharyngeal and gut microbiota-based that could aid predicting COVID-19 severity. The study reported significant changes in the richness and Shannon diversity index in nasopharyngeal microbiome associated with severe symptoms.

Strengths:

The study successfully identified differences in the microbiome diversity that could indicate or predict disease severity. Furthermore, the authors demonstrated a link between individual nasopharyngeal organisms and the severity of SARS-CoV-2 infection. The density of the nasopharyngeal organism was shown to be a potential predictors of severity of COVID-19.

---

## [Referee Report · Reviewer #3 (Public review)]

Summary:

How the microbial composition of the human body is influenced by and influences disease progression is an important topic. For people with COVID-19, symptomatic progression and deterioration can be difficult to predict. This manuscript attempts to associate the nasal and fecal microbiomes of COVID-19 patients with the severity of disease symptoms, with the goal of identifying microbial markers that can predict disease outcomes.

Strengths:

Analysis of microbiomes from two distinct anatomical locations and across three distinct patient groups is a substantial undertaking. How these microbiomes influence and are influenced by COVID-19 disease progression is an important question. In particular, the putative biomarker identified here could be of clinical value with additional research.

Weaknesses:

The primary weaknesses of this analysis is the relatively low sample size for analyzing disease subsets and moderate correlation values observed for putative biomarkers. Regardless, this data can be used to inform future studies aiming to understand the contribution of multifactorial dysbiosis to COVID-19 disease progression.

---

## [Author Response]

The following is the authors’ response to the original reviews.

**Public reviews:**

**Reviewer 1**

We would like to express our gratitude to Reviewer 1 for providing a thorough summary of our work and highlighting its strengths. With regards to the weaknesses, we are committed to improve the manuscript by performing the necessary changes. First, we will specify the exact p-value in all cases.

Regarding the discussion section, we acknowledge the feedback regarding its potential confusion. In line with the reviewer's suggestion, we will reduce the literature review and highlight our findings.

Finally, for the preprint we did not include cofounders such as HIV infection and ethnicity as our study population did not exhibit viral infections and comprised only Hispanic individuals. We will make a more thorough description of the population of study and address these characteristics explicitly in both the methods section and the initial part of the results.

**Reviewer 2**

We appreciate and thank reviewer 2 for the commentaries. Although it is true that several papers have described the role of microbiome in COVID-19 severity, we firmly believe that our current work stands out. There is not much information related to this association in Mediterranean countries, especially in the south of Spain. In addition, most of the studies only describe microbiota composition in stool or nasopharyngeal samples separately, without investigating any potential relationships between them as we do.

(1) We agree with the reviewer idea of a limited sample size. We faced the challenge of collecting the samples during the peak of COVID-19 pandemia. Thus, doctors and nurses were overwhelmed and not always available for carrying out patient recruitment following the inclusion criteria. Despite these constraints, we ensured that all included samples met our specified inclusion criteria and were from subjects with confirmed symptomatology.

In addition, our main goal was to identify whether severity of the disease could be assessed through microbiota composition. Therefore we did not include a healthy group. Despite not having a large N, our results should be reproducible as they are supported by statistical analysis.

(2) We thank reviewer commentary, and since our original sentence may have lacked clarity, we intend to modify it to ensure it conveys the intended meaning more effectively.

Nonetheless, we remain confident in the significance of our findings. Not only have we found correlation between microbiota and COVID severity, but we have also described how specific bacteria from each condition is associated with key biochemical parameters of clinical COVID infection.

(3) We appreciate the feedback provided by the reviewer. In this case, we have performed 16S analysis due to its cost-effectiveness compared to metagenomic approaches. Furthermore, 16S analysis has undergone refinements that ensure comprehensive coverage and depth, along with standardized analysis protocols. Unlike 16S, metagenomic approaches lack software tools such as QIIME that facilitate standardization of analysis and, thus, reduce reproducibility of results.

(4) We sincerely appreciate this insightful suggestion. simply listing associations between both microbiomes and COVID-19 severity could not be enough, we intend to discuss how microbiota composition may be linked to the mechanisms underlying COVID-19 pathogenesis in our discussion.

(5) We are grateful for the constructive criticism and intend to rewrite our abstract to enhance clarity. Additionally, we will thoroughly review all figures and their descriptions to ensure accuracy and comprehensibility.

**Reviewer 3**

We acknowledge the annotations made by reviewer 3 and are committed to addressing all identified weaknesses to enhance the quality of our work. Our idea is to modify the methods section and figures to make them easier to understand.

Specifically, in the case of Figure 1, we recognize an error in the description of the Bray-Curtis test. We appreciate the commentary and we will make the necessary changes. Moreover, there is another observation related to Figure 1 description. We are going to modify it in order to gain accuracy.

For figure 2 we are planning to add a supplementary table showing the abundance of detected genus. Nevermind, we will also update the manuscript text to provide clarification on how we obtained this result.

Regarding the clarification about "1% abundance," we want to emphasize that we are referring to relative abundance, where 1 represents 100%. To avoid confusion, we will explicitly state this in both the methods section and figure descriptions. Besides, it is true that the statistical test employed for the analysis is not mentioned in the figure description and we recognize that the image may be difficult to interpret. Therefore, we will modify the text and a supplementary table displaying the abundance and p values is going to be added.

Furthermore, we agree with the reviewer's suggestion to investigate whether the bacteria identified as potential biomarkers for each condition are specific to their respective severity index or if there is a threshold. Thus, we will reanalyze the data and include a supplementary table with the abundance of each biomarker for each condition. We will also place greater emphasis on these results in our discussion.

Finally, in response to the reviewer's suggestion, we are going to go through the nasopharyngeal-fecal axis part in the discussion. It is well described that COVID-19 induces a dysbiosis in both microbiomes. Consequently, we understand that the ratio we have described could be an interesting tool for assessing COVID severity development as it considers alterations in both environments. However, we acknowledge that there may be room for improvement in clarifying the significance of this intriguing finding and its implications.